# The Major Role of T Regulatory Cells in the Efficiency of Vaccination in General and Immunocompromised Populations: A Review

**DOI:** 10.3390/vaccines12090992

**Published:** 2024-08-30

**Authors:** Stanislaw Stepkowski, Dulat Bekbolsynov, Jared Oenick, Surina Brar, Beata Mierzejewska, Michael A. Rees, Obi Ekwenna

**Affiliations:** 1Department of Medical Microbiology and Immunology, University of Toledo, Toledo, OH 43614, USA; dulat.bekbolsynov@utoledo.edu (D.B.); beata.mierzejewska@rockets.utoledo.edu (B.M.); 2Neurological Surgery, The University of Toledo College of Medicine and Life Sciences, Toledo, OH 43614, USA; jared.oenick@rockets.utoledo.edu; 3Department of Urology, The University of Toledo College of Medicine and Life Sciences, Toledo, OH 43614, USA; michael.rees2@utoledo.edu (M.A.R.); obinna.ekwenna@utoledo.edu (O.E.)

**Keywords:** vaccines, T cells, T regulatory cells, immunocompromised, vaccine efficacy

## Abstract

Since their conception with the smallpox vaccine, vaccines used worldwide have mitigated multiple pandemics, including the recent COVID-19 outbreak. Insightful studies have uncovered the complexities of different functional networks of CD4 T cells (T helper 1 (Th1); Th2, Th17) and CD8 T cells (T cytotoxic; Tc), as well as B cell (B^IgM^, B^IgG^, B^IgA^ and B^IgE^) subsets, during the response to vaccination. Both T and B cell subsets form central, peripheral, and tissue-resident subsets during vaccination. It has also become apparent that each vaccination forms a network of T regulatory subsets, namely CD4+ CD25+ Foxp3^+^ T regulatory (Treg) cells and interleukin-10 (IL-10)-producing CD4+ Foxp3^−^ T regulatory 1 (Tr1), as well as many others, which shape the quality/quantity of vaccine-specific IgM, IgG, and IgA antibody production. These components are especially critical for immunocompromised patients, such as older individuals and allograft recipients, as their vaccination may be ineffective or less effective. This review focuses on considering how the pre- and post-vaccination Treg/Tr1 levels influence the vaccination efficacy. Experimental and clinical work has revealed that Treg/Tr1 involvement evokes different immune mechanisms in diminishing vaccine-induced cellular/humoral responses. Alternative steps may be considered to improve the vaccination response, such as increasing the dose, changing the delivery route, and/or repeated booster doses of vaccines. Vaccination may be combined with anti-CD25 (IL-2Rα chain) or anti-programmed cell death protein 1 (PD-1) monoclonal antibodies (mAb) to decrease the Tregs and boost the T/B cell immune response. All of these data and strategies for immunizations are presented and discussed, aiming to improve the efficacy of vaccination in humans and especially in immunocompromised and older individuals, as well as organ transplant patients.

## 1. Introduction

The concept of natural cowpox vaccination against human smallpox, discovered in 1796 by Edward Jenner [1], evolved 162 years later into a global vaccination campaign promoted by the World Health Organization (WHO), culminating in the eradication of smallpox between 1958 and 1977 [2]. The subsequent development of an attenuated virus vaccine against polio in 1950 [3] ultimately saved millions of lives, and this was followed by vaccines against measles in 1954 [4] and against different strains of influenza since 2003 [5,6,7], as well as remarkably successful mRNA-based vaccines against COVID-19 in 2019 [8]. Nowadays, vaccination against multiple viruses, bacteria, and parasites has become routine practice to protect humans from infection by building an “immune umbrella” of vaccine-specific memory T (Tm) and B (Bm) cells with circulating immunoglobin (Ig)M, IgG, and IgA antibodies [9,10,11].

Independently of the type of vaccine, the immune system mounts a complex response involving different functional networks of CD4 T cells with T helper 1 (Th1), Th2, and Th17 cells, as well as T follicular helper (Tfh) cells and CD8 T cytotoxic (CTL) cells and their respective Tm subsets. Simultaneously, vaccine-specific B cell clones expand into B^IgM^, B^IgG^, B^IgA^, and B^IgE^ cells and their Bm subsets, assisted by Tfh and other Th subsets [12]. These multiple Tm and Bm cells establish “trained” immunity [13], which is influenced by multiple factors (e.g., sex, age, preexisting conditions, etc.), as well as individual immunocompetence or exposure to immunosuppression, such as in transplant recipients [12,14].

In vaccine-protected individuals, an initial response via innate immunity is quickly supported by circulating vaccine-specific IgM/IgG/IgA antibodies [15]. Effective vaccination engages first vaccine-specific Th1/Tc/Th17/Tfh cells, assisting the production of IgM/IgG/IgA antibodies [16]. However, it has also become apparent that the network of potent T regulatory (Treg) subsets affects the efficacy of vaccination. These subsets include classical CD4^+^CD25^+^Foxp3^+^ Treg cells and CD4^+^Foxp3^−^ interleukin-10 (IL-10)-producing T regulatory 1 (Tr1), IL-35-producing Tr35, and the recently described transient Treg cells (see text below). Overall, these and other regulatory subsets actively shape the immune response, influencing the quality/quantity of the T/B cell response and IgM/IgG/IgA antibody production during the vaccination process [17].

## 2. General Description of Memory T Cell Immune Response during Vaccination

The most fundamental role in immunization with any vaccine is played by the effective generation of multiple Tm (memory) cells, including central (Tcm), effector (Tem), and tissue-resident Tm (Trm) cells [18] (Figure 1). These three subsets (Tcm, Tem, and Trm) have unique membrane markers and distinctive homing potentials [19] (Figure 2). Like naïve T cells, Tcm cells patrol secondary lymphoid organs (SLO) such as the lymph nodes (LNs) and the white pulp of the spleen, as both are guided by LN-homing CD62L, C-C-chemokine receptor type 7 (CCR7), and sphingosine-1-phosphate receptor-1 (S1pr1) molecules [20,21] (Figure 2A). In response to a vaccine (or other antigen), naïve T and Tcm cells expand by proliferation and differentiate into CD62L^−^CCR7^−^ T effector cells (Teff) and Teff memory cells (Tem); in this process, Teff/Tem cells change their homing receptors to migrate from the SLO to multiple tissue types, searching for infection sites to eliminate viruses/bacteria/parasite infections (Figure 2B). Like Teff cells, but much faster, Tem cells proliferate and expand into different functional subsets, including cytotoxic T (Tc) CTL cells [22,23]. After depleting LN-homing receptors CD62L and CCR7, Tem cells recirculate between the blood and non-lymphoid tissue (NLT). Upon extravasating into tissue, human Tem cells also lose CD45RA but retain CD25 (IL-2a receptor chain), CD45RO, and CD127 (IL-7α receptor chain), which are important in cell trafficking, as they enter the tissue through the blood and exit the tissue through afferent lymphatic vessels towards LNs [19] (Figure 2B).

While T cells express CCR7 to egress NLT, Trm cells remove CCR7 to permanently reside in tissue [18]. CD4^+^ Trm cells also decrease the expression of CCR7, CD62L, and S1pr1 but express CXCL10-homing signaling molecules [24]; Trm cells release CXCL10 chemokines to attract CXCR3 receptor-expressing CD4^+^/CD8^+^ functional T and other cells [25] (Figure 2C). The Tm immune umbrella is supported by networks of Bm cells producing vaccine-specific antibodies.

## 3. General Description of Different T Regulatory Cells

The concept of suppressive lymphocytes, suggested in the 1970s [26], was explored extensively for organ transplantation in the 1980s in rats as CD4^+^ and CD8^+^ cells [27]. The first specific marker was identified as an IL-2α receptor chain (CD25), described 13 years later in mice [28] and then in humans [29,30,31]. The same CD4^+^CD25^+^ Treg cells were identified to express a master regulator, named the forkhead box P3 (Foxp3) transcription factor [32,33]. As indicated in Table 1, the most stable CD4^+^CD25^+^Foxp3^+^ Tregs are produced in the thymus as thymus-derived Tregs (tTregs) or natural Tregs (nTregs).

Another population of CD4^+^CD25^+^Foxp3^+^ T cells develops in the non-thymic peripheral tissue as peripheral Tregs (pTregs). Tregs can be characterized based on their functional profile as central Tregs (cTregs) or effector Tregs (eTregs) [41]. Antigen stimulation induces cTregs to differentiate into eTregs, which are identified as either the CD62L^low^ CD44^high^ or CCR7^low^ CD44^high^ phenotype. eTregs also express higher levels of cytotoxic T-lymphocyte-associated protein 4 (CTLA-4) and inducible T cell co-stimulator (ICOS) molecules compared to cTregs [41,42]. Most eTregs migrate to NLT and especially to inflammatory NLT sites to downregulate the immune response and maintain an immune homeostatic balance among other T cells, thus contributing to tissue repair and regeneration [43,44]. Similarly to other T cell subsets, eTregs migrate to NLT by expressing homing chemokine receptors (CCR4, CCR6, and CCR10) and recirculation markers (CD62L and CCR7) [41,42]. These different Tregs, maintaining homeostasis in SLO and NLT areas, are depicted in Figure 2. A unique subset of inducible Treg (iTreg) cells is produced ex vivo by exposure to low-dose IL-2 and the polyclonal activation of TCRs [45,46] (Table 1). 

Different CD4^+^CD25^+^Foxp3^+^ Treg subsets have their own patterns, expressing Foxp3 master regulators, influencing their stability, and affecting their suppressive capacity [28]. Constant CD25 expression (IL-2Rα receptor chain) competes for free IL-2, raising the activation threshold for other T cells, creating an additional regulatory tool [47] (Figure 3A). Despite accounting for only 3–10% of all CD4^+^ T cells, Tregs maintain self-tolerance and homeostatic immune balance during the response to vaccines and other antigens. They secrete anti-inflammatory cytokines such as IL-10, IL-35, and TGF-β [48,49,50] (Table 1). As shown in Figure 4, contact-dependent suppression is mediated via co-stimulatory molecules, such as CTLA-4 binding to CTLA-4L on antigen-presenting cells, as well as programmed cell death protein 1 (PD-1) binding to the PD-1 ligand (PD-L1) [51,52,53]. Indeed, there are co-stimulatory and co-inhibitory molecules expressed on T cells and APCs, which are actively involved in regulating the immune response to vaccines. Thus, multiple regulatory mechanisms affect the vaccination efficacy as they originally were designed to prevent the occurrence of excessive immune responses, which result in autoimmune diseases [54,55]. 

In addition to tTreg [56], pTreg, and iTreg cells [57,58] (Figure 3A), there are Treg subsets that are transiently transformed from CD4^+^ Foxp3^+^ Tregs when they are operating in the neighborhood of different Th (T helper) subsets [59] (Figure 3B). Like conventional CD4^+^ T cells, Tregs (Figure 3A) in a Th1-polarized environment upregulate the Tbet transcription factor (TF) to become Tbet^+^Foxp3^+^ Th1-Tregs [60] (Figure 3B). In a similar fashion, the Th2 type drives the GATA3 TF in Tregs to induce Th2-Tregs; Th17 cytokines drive the RORγt TF in Tregs to switch to Th17-Tregs; and a Tfh environment induces the Bcl6 TF to convert them into Tfh-Tregs [38,61] (Figure 3B). Classical Tr1 cells are characterized as an IL-10-producing CD4^+^Foxp3^−^ phenotype subset with a potent suppressive function [39] (Figure 3C). Their regulatory power also relies on the production of IL-35 and TGF-β and the expression of downregulatory molecules such as lymphocyte activation gene 3 (LAG-3), ICOS, and PD-1, like Tregs [62]. Studies have also identified the eomesodermin (EOMES) TF-expressing IL-10/IFN-γ co-producing Tr1-like cells [39,63]. The transient IL-17-producing proinflammatory Th17 cells were identified as Th17-Tr1 cells switching into Tr1 cells with high levels of IL-10 [39,64]. Another regulatory subset of Tr35 cells produces suppressive IL-35-inhibiting dendritic cells (DCs) and T cells, including proinflammatory Th1 and Th17 cells [40] (Figure 3C). Furthermore, IL-35 induces CTLA4, ICOS, and PD-1 [65] and it promotes the proliferation of other Treg cells [40]. This very complex system of regulatory networks plays a critical role in shaping the vaccination environment. Although our review presents data about the role of Treg and Tr1 cells in vaccination, all of these subsets may be involved in the response to vaccines. 

## 4. Regulatory T Cells Downregulate the Immune Response to Vaccines

### 4.1. Pre-Vaccination Treg Level Affects Vaccination Efficacy

It is well established now that high pre-vaccination Tre/Tr1 levels affect the vaccination efficacy. This is especially observed in older people, who are recommended in Europe and United States to receive higher vaccine doses, novel adjuvants, and early vaccination schedules [66]. The most convincing results showed that elderly subjects, having higher levels of Tregs prior to influenza vaccination, were associated with lower immune responses (reviewed in [67]).

When individuals over 60 years old were vaccinated against the influenza virus, they were divided into responders and non-responders [68]. Non-responders showed a tendency for a higher frequency of Tregs, and the relative increase in Tregs was associated with a poor response to the influenza vaccine. However, it can be stated that the pre-vaccination Treg levels influence the vaccination efficacy not only in the elderly but also in other people, especially those with an immunocompromised immune status, such as organ transplant recipients.

During ageing, the thymus undergoes dramatic involution, resulting in only a fraction of functional thymic tissue remaining among people over 70 years old [69]. Experiments in aged mice showed that Tregs were disproportionately elevated in SLO [70,71]. This accumulation of Tregs shifted the balance towards downregulation, effectively lowering the possibility for a rapid immune response after vaccination (as reviewed in [67]). A subset of tTregs accumulated predominantly in the lymph nodes (LNs) of SLO in mice [72].

Similar conclusions about Tregs during ageing were reported in humans [73,74,75,76,77]. One analysis found 5.8 ± 0.4% Foxp3^+^ Tregs in the blood in older humans and 4.4 ± 0.4% Tregs in young persons (<30 years old; *p* = 0.03) [73]. Elderly people with elevated pre-vaccination levels of Tregs, as well as senescent T effector cells, are currently recommended to receive higher doses of vaccines against influenza, herpes zoster, and pneumococcal disease, as well as additional booster doses of vaccines against tetanus, diphtheria, and pertussis [78].

Similarly, the anti-herpes vaccine (Zostavax), which is a live-attenuated virus, was poorly effective in older adults because the Tregs reduced the efficacy of the vaccination in elderly subjects [67,79]. Since primary vaccine responses are mostly poor in older people, inducing diminished long-term protection, novel approaches are being considered to reduce the risk of infection (see section below regarding methods to improve vaccination). Table 2 depicts the effect of high Treg levels prior to and after vaccination on vaccination efficiency in normal individuals, older people, cancer patients, and immunocompromised individuals such as transplant recipients.

### 4.2. Presence of Tregs Affects Efficacy after Vaccination against Infectious Diseases

Emerging experimental and clinical data demonstrate that the presence of Treg, Tr1, and/or other regulatory T cells influences the vaccination outcomes regarding infectious diseases in children, adults, and especially older people [80] (Table 2). The role of Tregs was investigated regarding pneumococcal, tetanus, diphtheria, and other vaccines, concluding that Tregs played a critical role in shaping the immune response against vaccination [79]. In addition to adults and the elderly, the same review reported one study in infants wherein Tregs suppressed the antibody response to measles but not to the diphtheria–tetanus–pertussis (DTP) vaccine [79]. Thus, classical CD4^+^CD25^+^ Foxp3^+^ Tregs inhibit the immune response to several vaccines during and after vaccination [80]. Elaborate network of Tregs and Tr1-type cells have been described for coronavirus disease 2019 (COVID-19) vaccination with mRNA vaccines [81,82].

A live-attenuated *Streptococcus pneumoniae* (*S. pneumoniae*) vaccine (SPY1) induced Tregs, which downregulated the Th2/Th17 immune responses to the SPY1 vaccine. Interestingly, the Treg response was complex when the analysis was extended from the vaccination effect to post-vaccination challenge with *S. pneumoniae* infection. The protective effect produced by the SPY1 vaccine was modulated by the injection of a synthetic peptide (P17; representing *S. pneumoniae* antigen), which downregulated the Treg response. The P17 peptide treatment inhibited the increased levels of IL-10 and IL-6 in SPY1 vaccine-immunized mice [83]. Such treatment with the P17 peptide resulted in more severe pulmonary injury and worse survival, explained by the active participation of Tregs in alleviating *S. pneumoniae*-induced damage via a strong (possibly excessive) immune response. This experiment reveals the duality of the roles played by Tregs during vaccination vs. post-vaccination challenge with *Staphylococcus* infection. A detailed analysis revealed that the TGF-β-dependent Smad2/3 signaling pathways were actively involved [79,84]. SPY1 vaccination activated Foxp3, increasing the frequency of CD4^+^CD25^+^Foxp3^+^ Tregs during vaccination; SPY1 elevated Smad2/3 and phosphor-Smad2/3 downregulated a negative signal by Smad7 during live bacterial infection (Smad 1–9 regulate TGF-β receptor signaling) [85]. This complex signaling was reversed by the injection of the P17 peptide, revealing the opposite roles played by Tregs during vaccination compared to infection [84]. Thus, while a vaccine-induced immune response predicts the vaccine efficacy as modulated by Tregs, the exact role of Tregs during an infection seems to be complex and remains obscure. New approaches, such as proteomics, genomics, and transcriptomics, need to be explored to unravel at a single-cell level the complexity of regulation by T cells prior to, during, and after vaccination vs. infection [86].

Inactivated vaccines with chemically killed pathogens are safer and more stable than live-attenuated (LA) vaccines, but the latter produce better and longer-lasting protection [87]. Since inactivated vaccines induce weak immunity, they require repeated doses to generate a humoral response, which often is not permanent. The currently used inactivated vaccines are those against poliomyelitis (i.e., inactivated polio vaccine; IPV), hepatitis A, rabies, and influenza [87]. In contrast, LA vaccines are more potent in inducing long-lasting protection. For example, a temperature-sensitive NS1 gene-deleted LA influenza vaccine (DelNS1-LAIV) showed potent protective efficacy in the intranasal (i.n.) and intradermal (i.d.) vaccination of mice. DelNS1-LAIV i.d. vaccination conferred effective and long-lasting protection against a lethal virus challenge in mice, an effect that is impossible to achieve with an inactivated influenza vaccine. A single i.d. injection of DelNS1-LAIV produced 100% survival after an A(H1N1)09 influenza virus (H1N1/415742Md) challenge, suggesting a lack of negative interference by Treg/Tr1 cells [88]. In contrast, peptide-based influenza vaccines, which were poorly immunogenic, induced potent specific Tregs [89]. Thus, the type of vaccine influences the Treg/Tr1 immune response after vaccination.

Leishmaniasis (LZ), a parasitic *Leishmania major* (*L. major*) protozoa-caused disease, could be prevented in 90% of mice by a very low dose (not infectious) of intradermal (i.d.) inoculation with live *L. major* [90]. After *L. major* infection, the mice developed a specific Th1-type immune response, and it was significantly boosted by previous vaccination with killed parasites. Aged mice (>30 months old) showed a doubled number of Tregs in the LNs, while it was slightly elevated in the spleen and blood [90]. Tregs from old Foxp3-GFP knockout mice exhibited higher levels of suppression than those of young mice, which was associated with a poor response to *L. major* infection because the Tregs suppressed the IFN-γ-producing Th1 response. Consequently, the role of Tregs in *L. major* vs. *S. pneumoniae* infection is very different: they aggravated infections by parasites while protecting mice from the powerful lung damage caused by *S. pneumoniae*.

Several other works have revealed that the presence of Tregs reduces the vaccination efficacy regarding infectious agents [81,91,92]. A recent study showed how Tregs interfered with the COVID-19 mRNA vaccination’s efficiency [81]. The expansion of both Tcm and Tem cells was impeded after mRNA vaccination by CD4^+^CD25^high^ Treg cells expressing CCR6 chemokine receptors, which decreased the SARS-CoV-2 IgG production. Furthermore, booster mRNA vaccination did not affect the number of Tregs, persistently reducing the mRNA vaccination efficacy. Treg cells suppressed IFN-γ-producing SARS-CoV-2-specific Th1 cells.

Another analysis of post-vaccination Tregs against *Helicobacter pylori* (*H. pylori*) concluded that the reduction of Tregs improved the anti-*H. pylori* vaccination efficacy [91]. The immune response to *H. pylori* involved Th1, Th17, and Treg cells, while vaccination produced a significant reduction in the *H. pylori* colonization levels. Different strategies were adopted during vaccination with the consideration of boosting the Th1/Th17 response and minimizing the Treg response.

Research on influenza vaccination in mice showed that the expansion of CD4^+^Foxp3^+^ Tregs did not alter the primary B cell response but rather suppressed the primary and memory Th1 responses, which are vital for robust immune defense [93]. The study also showed that human Tregs may transiently express Foxp3 and be converted from CD25^−^ to CD25^+^ after homeostatic expansion, with high CD127 (IL-7α receptor) expression indicating an activation status. The role of Tregs in the immune response to influenza vaccination was also demonstrated to be mediated through the secretion of IL-10, IL-35, and TGF-β [93]. These findings suggest that Tregs modulate the immune environment by positively and negatively regulating the effectiveness of influenza vaccines. The variations in effective vaccination underline the role of multiple factors, but Treg-based regulation needs exploration as one of the main players, and it may be modified to achieve potent vaccination. The impact of the Treg/Tr1 balance under the dominance of regulation on vaccination outcomes is presented in Table 2. High Treg/Tr1 levels are correlated with a deficiency in vaccination efficacy, predominantly in older individuals and immunocompromised patients; normal Treg/Tr1 levels are also disadvantageous for immunosuppressed individuals, and low Treg/Tr1 levels need to be achieved by therapeutic intervention to decrease their regulation.

### 4.3. Presence of Tregs Affects Efficacy after Vaccination against Cancer

Therapeutic vaccination against cancer has been shown to exhibit broad prospects in individualized therapy [94]. Among other technological challenges, the neutralization of Tregs and other types of regulation has proven to be a necessary step for successful outcomes [94,95]. For example, one study highlighted a regulatory challenge in vaccination with a melanoma peptide or melanoma lysate loaded to antigen-presenting cells (APCs; cancer antigen/APC vaccine). Such vaccination produced a robust antigen-specific CTL response, reaching CTL peak expansion at day 7, but it quickly contracted at day 28, a disappointing failure in therapy [96]. This dramatic change was attributed to the apoptosis of CTLs but also to the needless expansion of IL-10-producing CD4^+^CD25^+^ Tregs. The difficulty in eliciting a strong CTL response against melanoma self-antigens is obviously underscored by the active peripheral tolerance maintained continuously by CD4^+^CD25^+^Foxp3^+^ cTregs/pTregs [96].

As previously documented, programmed cell death protein 1 (PD-1) expression impedes vaccination against tumor development [92]. Combined vaccination against tumors with the administration of anti-PD-1 mAb reduced the Tregs and enhanced the effector Tm cells. In cancer vaccination trials, the efficacy was affected by Tregs as selective Treg depletion was continuously associated with an enhanced antitumor response [86]. Research has shifted now towards identifying tumor antigens that are less affected by Treg suppression and to targeting Tregs with antibodies against different immune checkpoints, opening new avenues for enhanced vaccine efficacy [86].

Extensive studies have been performed using an in vivo Foxp3^+^ Treg depletion system in the bacterial artificial chromosome (BAC)-transgenic depletion of regulatory T cells (DEREG) mouse model [97]. DEREG mice are a useful model in Treg research as they express a transgenic diphtheria toxin receptor under the control of the *Foxp3* locus, thus allowing for the selective depletion of Foxp3-expressing Tregs. The depletion of Tregs in DEREG mice results in the partial regression of ovalbumin (OVA)-expressing B16 melanoma cells, along with an increased intratumor level of activated CD8^+^ cytotoxic T cells (CTL) [98]. The antitumor effect was enhanced by Treg depletion after therapeutic vaccination against OVA complemented by selective Treg elimination, where vaccination against a self-antigen-bearing tumor evoked the most effective antitumor response [98]. Furthermore, crossing DEREG mice with RipOVA^low^ mice (expressing OVA neo-self-antigen under control of the rat insulin promoter) resulted in DEREG/RipOVA^low^ mice displaying the capacity for both Treg-selective depletion and OVA vaccination induced against self-antigens with potent tumor regression. Such sophisticated targeting of Tregs, combined with boosted vaccination, promises more efficient antitumor responses [98]. This approach overcomes the limitations of CD25 mAb therapy, which may deplete not only Tregs but also activated CD25^+^ Tem/Tcm cells. Tumor-bearing individuals had elevated numbers of CD4^+^CD25^+^Foxp3^+^ Tregs, which made it exceptionally difficult to mount efficient antitumor immune responses [98]. Multiple experimental and clinical efforts have explored the elimination of Tregs by depletion, with very promising results [99].

While the continuous development of complex vaccination strategies of Treg depletion with booster anticancer vaccines is the priority, anti-CD25 mAb therapy remains a viable option, as seen in multiple clinical trials [100]. However, an anti-CD25 mAb therapy was also optimized to deplete Tregs but preserve the IL-2/Stat5 signaling of Teff (T effector) cells. It is believed that Treg depletion with adjuvants represents a new strategy for improved therapeutic vaccination [80]. An effective anti-PD-1 mAb treatment should be enriched with other anti-Treg approaches and the additional boosting of the immune response with potent therapeutic vaccines.

Although there are several differences between solid tumors and leukemias in mature and elderly individuals and children, all of these patients are affected by the Treg/Tr1 functions during vaccination, including cancer vaccines. A recent approach to solid tumors investigated intratumor metabolic pathways. For example, the solid tumor immune microenvironment of colorectal cancer (CRC)—one of the most prevalent cancers—has prompted the exploration of CRCs with abnormal glycolysis, glutaminolysis, and increased lipid synthesis [101]. In addition, a platelet glycoprotein 4 (CD36) inhibitor combined with anti-PD-1 treatment increased the efficacy of intratumor Tregs [102,103], while a phospholipase A2 (cPLA2)-alpha inhibitor prevented the dysfunction and senescence of Teff cells for solid tumors in older subjects [104]. Although Tregs suppress protective host immune responses to solid tumors, their targeting is often limited to modulation signals (e.g., anti-CTLA-4 mAb). The ubiquitin-specific protease Usp7 was shown to control Tregs by stabilizing Foxp3. Since targeting Usp7 impaired Foxp3^+^ Tregs, antitumor vaccines were more effective with anti-Usp7 therapy combined with anti-PD1 mAb in mice [105]. 

### 4.4. Depletion of Treg Cells Improves Vaccination Efficacy

Several experimental studies and ongoing clinical trials have shown that the elimination of Tregs improves the vaccination efficacy. For example, Tregs and TGF-β downregulated the anti-influenza antibody response post-vaccination in mice. The depletion of Tregs by anti-CD25 mAb increased the anti-influenza-specific IgG antibody levels and shifted the immune profile towards a protective Th1 response; these adjustments produced effective protection against lethal influenza infection in mice [79].

There are other strategies that have been explored, such as blocking Treg functions with anti-CTLA-4Ig, as well as anti-PD1 or anti-CD39 mAb (all markers expressed on Treg cells), which restored the immune response against live-attenuated varicella-zoster virus (VZV; zoster herpes vaccine; Zostavax) vaccination. The reactivated VZV virus, which is responsible for chickenpox, causes a painful rash with blisters followed by chronic pain (shingles). The same work showed that the addition of Shingrix, an immune-adjuvanted VZV glycoprotein E subunit (gE), was highly effective in overpowering the inhibitory effects of Tregs in mice vaccinated with Zostavax [79,106].

A pneumococcal vaccine (PCV) conjugated to increase its efficacy resulted in the elevation of Tregs in lung-draining lymph nodes, the lungs, and the spleen. These Tregs suppressed Teff (T effector) cells in response to PCV. Treatment with anti-CD25 mAb reversed the suppression, confirming that Tregs actively modulated the immune response to PCV [79].

Elderly humans over 65 years old are at increased risk of influenza virus infection compared to younger individuals [107]. The difference in risk persists even for fully vaccinated elderly people. However, the depletion of Tregs using anti-CD25 mAbs resulted in an increase in the antibody response in aged mice [107]. Furthermore, the IgG and Th profiles were changed by Treg blockage. Aged mice that were treated with anti-CD25 mAbs prior to vaccination for Treg depletion were also more effectively protected against a lethal influenza virus challenge [107].

Peptide-based influenza-specific vaccines targeting the conserved epitopes of influenza virus provided protection against different influenza strains, but generally they were poorly immunogenic [89,108]. These sub-immunogenic conditions induced potent specific Tregs, which were further expanded by repeated vaccination with unadjuvanted influenza peptides. While the depletion of vaccine-induced antigen-specific Tregs promoted influenza viral clearance in vivo, the promotion of Tregs by weakly immunogenic peptides created significant challenges. To overcome such problems, the investigators immunized mice with a short single-stranded cytosine phosphoguanine (CpG)-adjuvanted influenza peptide via subcutaneous (SC) injection with an intranasal booster dose, restricting the recruitment of peptide-specific Tregs in the lungs while stimulating robust T cell immunity. The in vivo immunization of aged mice with SC injection, followed by intranasal boosting with CpG-adjuvanted peptides or whole-inactivated influenza vaccines, fully protected the mice from influenza virus infection [89]. The CpG-adjuvanted peptide vaccines thus provided heterosubtypic influenza protection (for RR8 strain with H1N1 surface protein and HKx31 strain with H3N2 surface protein) [109] by inhibiting Treg development and enhancing the T cell immunity [89]. The universal influenza vaccine candidates include Multimeric-001 (M-001) with nine peptides for conserved immunogenic epitopes [108]. In human trials, the M-001 vaccine induced robust humoral and cellular responses, which were further augmented with a Montanide ISA 51 VG adjuvant.

Immunity to herpes simplex virus (HSV) vaccines was significantly elevated when the Treg response was curtailed. The removal of CD25^+^ Treg cells using anti-CD25 mAbs prior to immunization with the CpG-enriched SSIEFARL peptide (H-2K^b^ envelope) or glycoprotein B (gB)–DNA immunization significantly enhanced the CD8^+^ CTL response to the immunodominant SSIEFARL peptide [110]. Treg-depleted mice had two- to threefold enhanced CD8^+^ T cell reactivity, which was evident in the acute and memory stages. The depletion of CD25^+^ Tregs during the memory response substantially increased the CD8^+^ Tm memory pool. The CD8^+^ Tm cells, generated with plasmid DNA in the absence of Tregs, cleared the virus compared with the controls [111]. The authors concluded that the CD25^+^ Tregs had both quantitative and qualitative effects on the CD8^+^ Tm cells generated by vaccination against HSV with gB DNA.

Other evidence supports CD25^+^ Tregs’ ability to suppress the priming and/or expansion of antigen-specific CD8^+^ T cells during DNA vaccine immunization, as well as the enhancement of the peak CD8^+^ Tm response after depleting Tregs [112]. Further studies found that Tregs were involved in the contraction phase of CD8^+^ Tm cells and affected the quality of the Tm pools. These authors suggested that the elimination of Tregs, or at least their temporary inhibition during vaccination, was important to control HBV infection via HBV-specific CTL responses in chronically infected individuals. The depletion of CD25^+^CD4^+^Foxp3^+^ Tregs enhanced the HBV-specific CD8^+^ Tm/CTL response primed by DNA vaccination [112].

Following SC immunization with the Bacillus Calmette–Guerin (BCG) vaccine against infection with *Mycobacterium tuberculosis* (*M. tuberculosis*), a study examined whether the recruitment/expansion of Tregs—concomitant with the induction of anti-BCG T cell responses—would limit the efficiency of the BCG vaccine in mice. Four weeks following BCG vaccination, the percentage of CD4^+^CD25^+^Foxp3^+^ Tregs within CD4^+^ cells remained constant, whereas the total number of Tregs increased proportionally to the CD4^+^ T cell expansion in the draining inguinal LNs [113]. A similar pattern with Tregs was observed in the lungs after *M. tuberculosis* was delivered by aerosol to mice: an infection with 1 x 10^4^ colony-forming units (CFU) of BCG or ≈100 CFU of *M. tuberculosis* by nasal delivery led to the expansion of Tregs proportional to other T cell subsets [114]. Effective vaccination may require the attenuation of Tregs during vaccination, especially in immunocompromised subjects [113].

Another work showed that Tregs restricted the vaccine-induced T cell responses in mice [115]. DEREG mice (diphtheria toxin (DT)-depleted Foxp3) were vaccinated with a recombinant *Bordetella adenylate* cyclase toxoid fused with an MHC class I-restricted epitope of the circumsporozoite protein (ACT-CSP) of *Plasmodium berghei* (Pb), a parasite causing malaria in rodents. The ACT-CSP construct introduced the CD8^+^ epitope of Pb-CSP into the MHC class I presentation pathway of APCs. This elaborate system demonstrated that the number of CSP-specific Tm cells increased only when Tregs were depleted in DEREG mice [115]. Such depletion of Tregs augmented the CSP-specific T effector cells against malaria.

An evaluation of Foxp3^+^ Tregs during the tick-borne encephalitis (TBE)-induced response showed that high responders had decreased Tregs [116]. In contrast, non-responders had selectively expanded Foxp3^+^ Tregs after a booster TBE dose. An analysis showed that IL-10-producing Tr1 cells were induced upon TBE stimulation [17]. In TBE non-responders, the expansion of TBE-specific Tregs [117] was responsible for an increased pool of Foxp3^+^ Tregs. The same work showed that high TGF-β levels also induced iTregs.

Another group observed the impact of expanded Tregs in non-responders to a hepatitis B vaccine (HBV) [118,119]; these investigators showed higher numbers of Foxp3^+^ Tregs following an HBV booster dose. Interestingly, the expansion/induction of Foxp3^+^ Tregs among HBV non-responders occurred upon vaccination with other vaccines, suggesting the induction of iTregs in the presence of TGF-β, rather than the expansion of HBV-specific Tregs after a booster HBV dose. These authors argued that HBV-specific Tregs occurred during earlier HBV vaccinations; unspecific stimulation by other vaccines may have resulted in decreased immune responses. Furthermore, the authors indicated that the high IL-10 levels may be attributed not only to Tregs and Tr1 cells but also to IL-10-producing B cells, which have been documented in multiple sclerosis [120,121] and systemic lupus erythematosus [122,123] and in the natural tolerance of renal transplant recipients [124].

Human CD19^+^CD24^high^CD38^high^ immature B cells displayed regulation via IL-10 induced by anti-CD40 stimulation [122]. In vitro, the IL-10-producing immature B cells suppressed the proliferative response of HBV-specific CD8^+^ T cells [125]. These B regulatory cells expanded the induction of FoxP3^+^ Tregs [126], further complicating the regulation after vaccination.

Similarly to other vaccines, therapeutic vaccination against HIV-1 increased the frequency of CD4^+^CD25^+^Foxp3^+^ Tregs [127]. The Tregs’ appearance thus marked an enhancement in the HIV-1-specific CD8^+^ CTL response. The vaccination of HIV-1 patients with two doses of autologous DCs loaded with HIV-1 peptides, undergoing antiretroviral therapy (*N* = 17) in a phase I therapeutic vaccine trial, demonstrated an increased blood frequency of CD4^+^CD25^high^Foxp3^+^ Tregs from 0.74% to 1.2% (12/17; *p* = 0.06). Therapeutic immunization with a DC/HIV-1 vaccine modestly increased Tregs while significantly increasing HIV-1-specific CD8^+^ CTLs [127]. Thus, the Tregs’ suppressive effect masked an increased vaccine-induced anti-HIV-1-specific response.

These results, as examples of vaccination against viruses, bacteria, and protozoa, show that the elimination of Tregs improves the vaccination efficacy. This may become a tool to improve the outcomes of vaccine-induced T cell responses and IgG/IgA antibody production in humans requiring enhanced vaccination procedures because of their compromised immune systems.

## 5. Role of Tr1 Cells in Vaccination of Transplant Recipients

The Tr1 cell subset was shown to regulate the immune response [39,128], including the response during the vaccination process [82]. Our analysis suggested that the Tr1 subset represents one of the regulatory mechanisms affecting both humoral and cellular responses in kidney transplant (KT) patients (Figure 5). The frequency of IL-10-producing Tr1 (fT_IL-10_) was measured in KT recipients compared to healthy volunteers after vaccination against COVID-19 (Figure 5A). All patients were vaccinated from months to years after receiving KTs. When the recipients were vaccinated with the Moderna vaccine, they displayed an almost four times lower fT_IL-10_ (16.9/5 × 10^4^) than the fT_IL-10_ observed in recipients vaccinated with the Pfizer-BioNTech vaccine (*p* = 0.016; Figure 5B). This suggested the dominant regulation of Tr1 cells, as our analysis showed that the Th1-induced response led to lower levels of fT_IFNg_, fT_TNFa_, and fT_IL-2_ cells (all *p* < 0.05), as well as significantly lower anti-SARS-CoV-2 IgG production (*p* < 0.05), in Pfizer-BioNTech-immunized patients. Although the frequency of T_TNF-α_ cells was independent of gender and age (but not race), the lower fT_IL-10_ was correlated with higher anti-SARS-CoV-2 IgG levels (*p* = 0.046; Figure 5C). A linear regression analysis showed that the increased Th1 levels correlated with higher IgG levels (*p* < 0.001) and that the fT_TNF-α_/fT_IL-10_ ratios reflected the IgG levels (Figure 5D). This reciprocal interplay of Th1 and Tr1 cells influenced IgG production, confirming the major role of regulation in COVID-19 vaccination. We proposed that the Tr1 subset was critical in the regulation of the vaccine-induced immune response. The efficacy of vaccination most likely depends on Tr1 and other T regulatory cells. Furthermore, regulation may influence the vaccination efficacy significantly in all humans, but especially in immunosuppressed patients [82]. 

## 6. Methods to Overcome Treg/Tr1 to Improve Vaccination Efficacy

The following different strategies may be considered to overcome the extensive Treg/Tr1 function and to improve vaccines’ effectiveness in normal subjects, but especially in immunocompromised and older people, whose immune responses are deficient for multiple reasons: (1) Treg/Tr1 elimination/inhibition; (2) higher vaccine doses; (3) alternative administration routes; (4) novel adjuvants; (5) immunomodulatory drugs; and (6) selective senolytic drugs to induce apoptosis in senescent cells without harming healthy cells in the elderly population [78,129,130].

### 6.1. Treg/Tr1 Modulation to Improve Vaccine Immunogenicity

Some strategies are based on the temporary depletion of Treg/Tr1 subsets [131,132]. As described in detail in our review, Treg and Tr1 cells diminish the vaccination effects, and the reduction of regulation significantly improves the efficacy of vaccination. Most work and ongoing clinical trials on Treg elimination focus on anticancer vaccines [133]. Almost 30 years of development in cancer vaccines has resulted in 360 currently active clinical trials with peptide-, mRNA-, DNA-, and APC-based vaccines. To eliminate/diminish Treg cells, antibodies are used to target CD40 (found on APCs and binding to CD40L, predominantly expressed on T cells), 4-1BB (CD137 or TNF receptor superfamily 9, TNFRSF9), OX40 (expressed on memory and effector T cells), and PD-1 (inhibitory receptor binding to PD-L1) molecules [134,135,136,137,138]. For example, the four-day administration of Daclizumab (anti-CD25 mAb; anti-IL-2a receptor chain) depleted CD4^+^CD25^+^Foxp3^+^ Tregs in the peripheral blood [139]. The temporary elimination/inhibition of Tregs may provide substantial benefits during vaccination in individuals prone to excessive immunosuppression and/or a weak immune response.

### 6.2. Other Methods to Improve Vaccination’s Efficacy and Overcome Strong Treg/Tr1 Function

The simplest approach to increasing the efficacy of vaccination is to increase the doses of vaccines and the number of boosting doses [140,141,142,143,144]. It became obvious during COVID-19 that kidney transplant recipients (KTR) needed to receive a third and fourth boosting dose of the mRNA vaccine to improve both the anti-SARS-CoV-2 IgG antibody concentration and percentage of protected transplant patients [144]. While most vaccines are delivered through intramuscular (IM) injection, the intradermal (ID) delivery of the vaccine was shown to produce a better immune response to the influenza vaccine [145]. Another option to improve the vaccination efficacy is the use of adjuvants, which enhance the immunogenic effects of antigens, including vaccines [146]. Effective adjuvants may substantially change the efficacy of vaccines for older and immunocompromised individuals, including immunosuppressed organ transplant patients.

Immunomodulatory drugs may be used to improve the efficacy of some vaccines [147]. Advances in vaccinology have revealed common antigenic epitopes for influenza viruses, but also the role of immunomodulatory agents such as signaling pathway inhibitors, which are undergoing preclinical development. The emergence of pandemics due to H1N1 influenza in 2009 and avian H7N9 influenza in 2013, as well as human cases of avian H5N1 influenza, indicates the need for new approaches, such as searching for common epitopes and immunomodulatory drugs. A promising source of immunomodulators is natural plant peptides, such as defensins, displaying also antiviral activity and the potential for synthetic formulation [148].

The greatest deficiency in vaccination is observed in the elderly population [149]. The addition of senolytic drugs induces the apoptosis of harmful senescent cells in older individuals without harming healthy cells [149]. Dasatinib, an ATP-competitive protein tyrosine kinase inhibitor, eliminated senescent human and mouse endothelial cells, and, when administered for 5 days to old mice, improved their heart function; moreover, following exercise, it maintained better heart capacity for 7 months [150].

Our review aimed at emphasizing the fundamental role of regulation in vaccination processes and their multifactorial aspects. Without a doubt, vaccination remains crucial in preventive medicine, with a worldwide impact on global health, as recently demonstrated during the COVID-19 pandemic. The quality of vaccination depends on many aspects but one of the most obvious is the health status of those being vaccinated. Healthy individuals are just a fraction of those who must be vaccinated for the strategy to work, justifying the search for innovative ways to maximize the preventive effects of multiple vaccines. Our understanding of the role of Treg/Tr1 cells and their regulation in general during vaccination remains crucial in terms of strategies to improve the vaccination efficacy. 

## 7. Summary

The efficacy of vaccination depends on the function of multiple vaccine-induced Tm/Bm cells and the active production/levels of vaccine/virus-specific IgM/IgG/IgA antibodies. The responses of naïve and memory T cells and B cells are actively regulated by multiple Treg subsets. The following points about the regulation of vaccination have already been established: (1) the pre-vaccination levels of Tregs are associated with the efficacy of vaccination, especially in the older population; (2) the post-vaccination levels of cTreg, pTreg, and Tr1 cells influence the levels of Th1 and Th17 cells; (3) the reduction of Tregs by targeting Foxp3 in mice or the reduction of Tregs with anti-CD25 mAbs or anti-PD-1 mAbs significantly increase the levels of vaccine-specific Th1, Th2, and Th17 subsets, as well as the levels of vaccine-specific IgG and IgA antibodies; (4) patients who have a compromised immune response due to immunosuppressive therapy display particularly strong effects of Tr1 suppression; (5) several experimental and clinical protocols that reduce Treg regulation are currently being tested to improve vaccination; and (6) multiple methods may be used to boost the vaccination efficacy in older people and immunocompromised patients, including immunosuppressed organ transplant recipients. As depicted in Table 2, high Treg/Tr1 levels prior to and after vaccination in normal individuals, older people, cancer patients, and immunocompromised individuals diminish the vaccination efficacy. Normal subjects with balanced Treg/Tr1 levels have excellent vaccination efficacy, while cancer patients, older people, and immunocompromised individuals require the reduction of their Treg/Tr1 levels to maximize the vaccination effect. Furthermore, low Treg/Tr1 levels, accomplished by therapeutic elimination or their blockage or suppression, significantly improve the effect of vaccination. 

## Figures and Tables

**Figure 1 vaccines-12-00992-f001:**
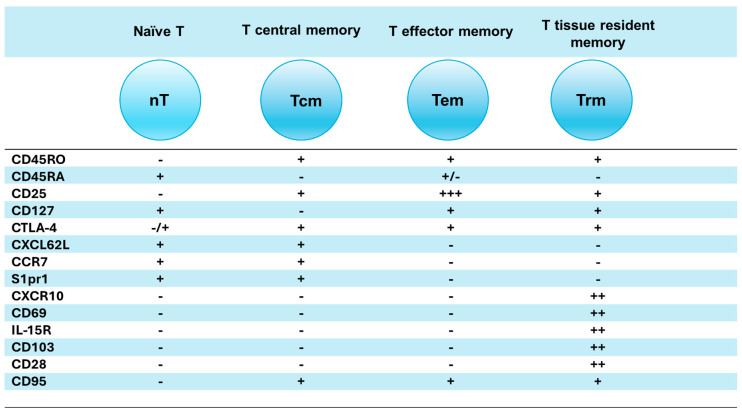
Representation of different T memory (Tm) subsets and their markers. Central Tm (Tcm) cells express CD45RA, CD25, CXCL62L, CCR7, and S1pr1 molecules; effector Tm cells (Tem) express CD45RO, CD25^high^, and CD127; and tissue-resident Tm cells (Tr) express CD45RO, CD25, CD127, CXCR10^high^, CD69^high^, IL-15Ra^high^ (receptor), and CD28^high^. The expression of each marker is shown with a single ‘+’ sign or more if highly positive, while negativity is shown with ‘−’.

**Figure 2 vaccines-12-00992-f002:**
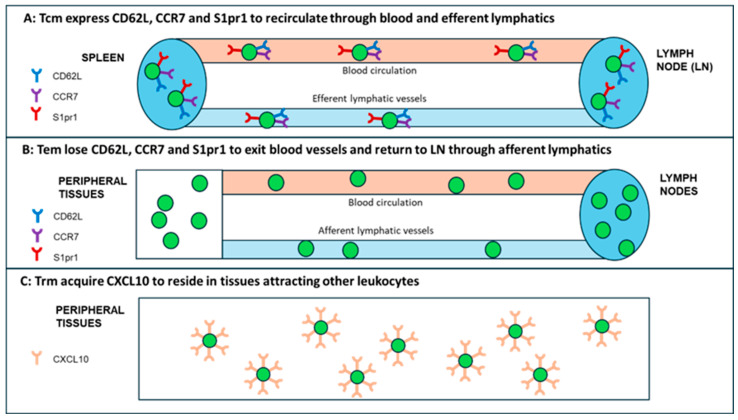
Memory T cell subsets with their distinct patterns of trafficking. (**A**) Central memory T (Tcm) cells recirculate between secondary lymphoid organs (SLO), which include lymph nodes (LNs) and spleen white pulp, using blood circulation and efferent lymphatic vessels. Tcm cells express CD62L, CCR7, and S1pr1 recirculation molecules. (**B**) Effector memory T cells (Tem) perform trafficking between tissue and LNs, exiting from the blood stream and returning through afferent lymphatic vessels. Tem cells downregulate CD62L, CCR7, and S1pr1 recirculation molecules. (**C**) Tissue-resident memory T (Trm) cells reside in tissue. Trm cells lose CD62L, CCR7, and S1sp1 and express CXCL10 receptors to attract CD4, CD8 T, and other cells expressing CCR3 molecules.

**Figure 3 vaccines-12-00992-f003:**
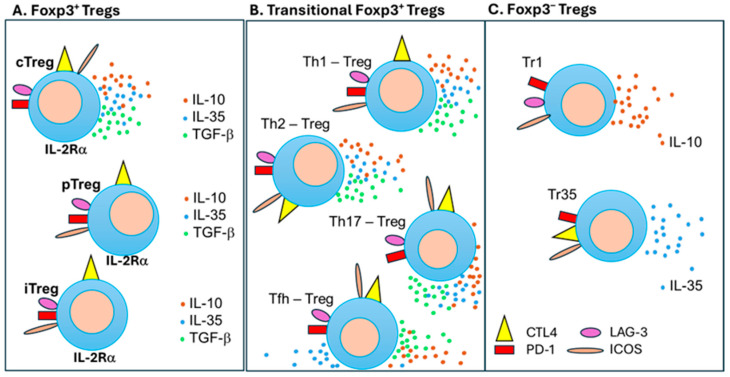
Diversity of Treg subsets involved in the regulation of the immune response to vaccines. (**A**) Foxp3-expressing Tregs are divided into central thymus-derived Tregs (cTregs), peripheral Foxp3-expressing Tregs (pTregs), and Foxp3-expressing induced Tregs (iTregs). (**B**) Transitional Foxp3-expressing Tregs emerge when Tregs function in a Th1, Th2, Th17, or Tfh environment: Tregs are transformed to Th1-Treg, Th2-Treg, Th17-Treg, and Tfh-Treg cells when they are exposed to a lineage-specific cytokine environment. (**C**) Foxp3^−^ IL-10-producing Tr1 cells and IL-35-producing Tr35 cells are potent regulators of the immune response. All of these T regulatory subsets express unique regulatory molecules (CTLA, ICOS, LAG-3, and PD-1), which bind to their respective ligands on APCs. Cytokines are marked as red dots for IL-10, blue dots for IL-35 and green dots for TGF-β.

**Figure 4 vaccines-12-00992-f004:**
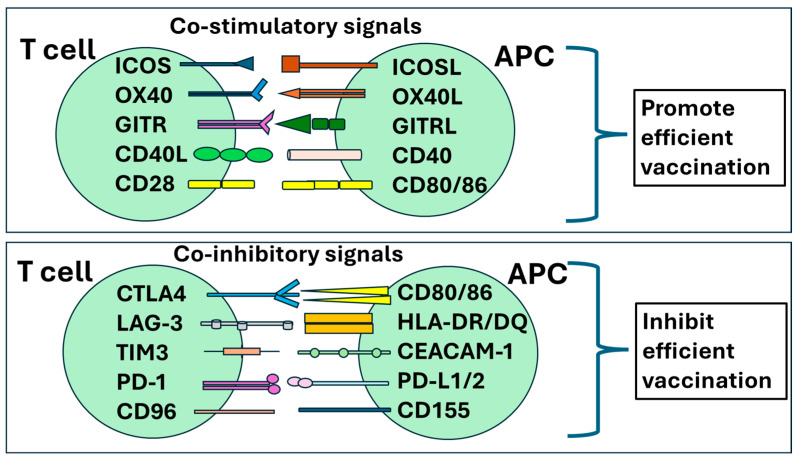
Multiple co-stimulatory and co-regulatory receptors on T cells and their ligands on APCs. Activated T cells recognize antigen-associated HLA complexes: the response is controlled by signaling through co-stimulatory and/or co-inhibitory molecules. Upper panel presents co-stimulatory molecules on T/APC cells: inducible costimulatory molecule (ICOS)/ICOSL; TNF receptor superfamily member 4 (OX40)/OX40L; glucocorticoid-induced TNF receptor family-related protein (GITR)/GITRL; CD40L/CD40; and CD28/CD80 or CD86. Lower panel presents co-inhibitory signals on T/APC: cytotoxic T-lymphocyte-associated antigen-4 (CTLA-4)/CD80 and CD86; lymphocyte activation gene-3 (LAG-3)/major histocompatibility complex II (MHC II); T cell immunoglobulin and mucin domain 3 (TIM-3)/carcinoembryonic antigen cell adhesion molecule 1 (CEACAM-1); programmed cell death protein-1 (PD-1)/PD-L1 or PD-L2; CD96/CD155.

**Figure 5 vaccines-12-00992-f005:**
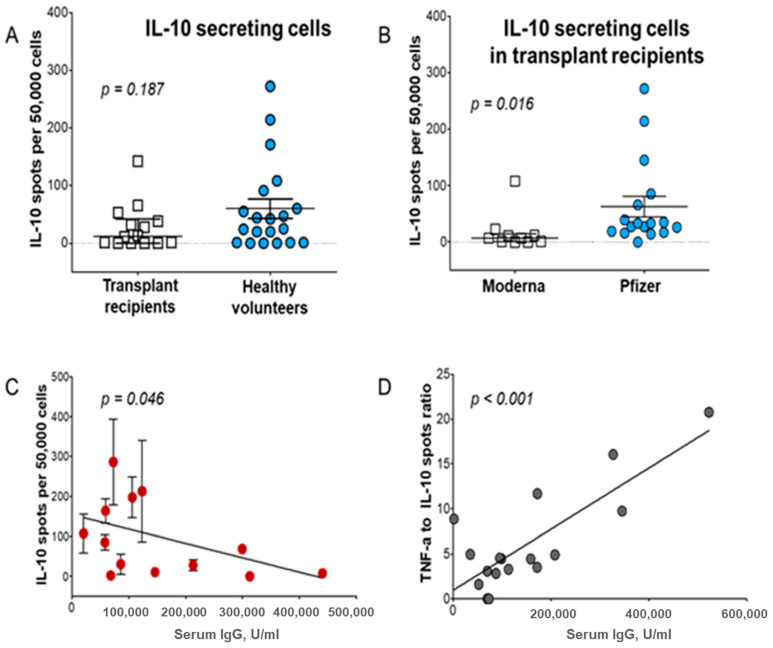
Evidence that the balance between the Th1 and Tr1 subsets regulates the anti-SARS-CoV-2 IgG titer after vaccination with mRNA anti-COVID-19 vaccines. PBMCs were harvested from kidney transplant recipients (KT) and healthy volunteers (HC) vaccinated against COVID-19 and cytokine secretion, measured by spot numbers in an ELISpot assay. (**A**) IL-10 secretion levels in transplant recipients compared to healthy volunteers; (**B**) KT recipients vaccinated with mRNA-1273 (Moderna) produced fewer ELISpots than peripheral blood mononuclear cells (PBMCs) from KT recipients vaccinated with the Pfizer-BioNTech vaccine. (**C**) Correlation between IL-10 secretion and antiviral antibody production. (**D**) Ratio of TNF-α production to IL-10 production directly correlates with vaccine efficiency, measured as antiviral IgG levels.

**Table 1 vaccines-12-00992-t001:** Functioning T regulatory cells and their markers.

T Regulatory Cells	Origin	Markers	References
tTreg	Thymus (central)	CCR7, CD45RA, CD31, SELL, NRP1, Foxp3+	Luo, Xue, Wang et al., 2019 [34]
pTreg	Peripheral	GATA3, IRF4, RORC, TBX21, HELIOS, Foxp3+	Zheng, Wang, Horwitz et al., 2008 [35]
iTreg	Peripheral	Treg-specific demethylation region (TSDR), Foxp3+	Zheng, Wang, Horwitz et al., 2008 [35]
Th1-Treg	Peripheral	CXCR, Tbet, Foxp3+	Halim et al., 2017 [36]
Th2-Treg	Peripheral	IL-4, IL-13, IRF4, Foxp3+	Halim et al., 2017 [36]
Th17-Treg	Peripheral	CD161, CCR6+, IL17A, IL23R, IL-12Rβ2, Fox3p+	Romagnani et al., 2009 [37]
Tfh-Treg	Peripheral	CXCR5, Bcl-6, IL-21, Foxp3+	Chung et al., 2011 [38]
Tr1	Peripheral	IL-10, CD49b, Lag3, Foxp3–	Suhrkamp et al., 2023 [39]
Tr35	Peripheral	IL-35, IL-10, p35, EB13, IL-12b2, gp130, Foxp3-	Yang, Dong, Zhong et al., 2024 [40]

**Table 2 vaccines-12-00992-t002:** Effects of Treg/Tr1 cells ratio on vaccination efficiency in various groups of individuals.

Group of Individuals	High Treg/Tr1 Ratio	Normal Treg/Tr1 Ratio	Low Treg/Tr1 Ratio
Prior to vaccination: healthy individuals	Good/deficient vaccination effect	Excellent vaccination effect	Excellent/good vaccination effect
After vaccination: healthy individuals	Good/deficient vaccination effect	Excellent/good vaccination effect	Excellent/good/deficient vaccination effect
After vaccination: healthy older individuals (>60 years-old)	Lack of/deficient vaccination effect; require higher and repeated doses ± adjuvant	Lack of/deficient/good vaccination effect; require higher and repeated doses ± adjuvant	Deficient/good vaccination effect; require higher and repeated doses ± adjuvant
After vaccination: cancer patients	Require anti-PD-1 mAb/addition of adjuvant + active removal of Tregs	Require anti-PD-1 mAb/addition of adjuvant ± active removal of Tregs	Require anti-PD-1 mAb/addition of adjuvant
After vaccination: transplant recipients	Lack of/deficient vaccination effect; require higher and repeated doses ± adjuvant	Lack of/deficient vaccination effect; require higher and repeated doses ± adjuvant	Lack of/deficient/good vaccination effect; require higher and repeated doses ±adjuvant

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
