# Peer review of "The Major Role of T Regulatory Cells in the Efficiency of Vaccination in General and Immunocompromised Populations: A Review"

_vaccines, 2024, doi:10.3390/vaccines12090992_

Round 1
Reviewer 1 Report
Comments and Suggestions for Authors
The submitted review by Stepkowski et al om the major role of T regulatory cells in efficiency of vaccination in general and immunocompromised populations addresses a complex and relevant topic in vaccine immunity. Overall, I found this to be a useful and well-organized review, with a complex topic presented in a simple and easy to understand manner. The authors cite most relevant work in the field and have included several figures and tables to summarize key information. I believe this review will be useful for readers looking to understand the role of Tregs in vaccine efficiency. Some suggestions are provided to improve the manuscript.
Major
1. L314-318: The comparison of the role of Tregs in vaccination vs infection is verry interesting. The authors can consider including some additional discussion about the status of similar differences in Treg function between live attenuated and inactivated approaches for vaccination (e.g. LAIV versus inactivated flu vaccines).
2. L587: some additional discussion on the effect of intranasal administration on T cell responses to vaccines will be useful as this is an area where several papers have been published comparing IN with other routes of vaccine delivery.
Minor:
1. L34: should read as ‘subsequent discovery of attenuated virus vaccine against polio”
2. L61: change ‘plaid’ to ‘played’
3. L67-68: there is a duplication of the sentence that needs to be corrected
4. Figure 1: the legend should clarify the difference between the combinations of single and double – and + signs used to denote phenotypes.
5. L90: should this read as ‘efferent’ instead of afferent?
6. L150-155: as written these sentences are unclear to me – consider rephrasing to improve clarity
7. L170: change ‘Th1-plarized’ (to polarized?)
8. Section 3.4: some initial information defining the Tr35 subset more rigorously would be useful here.
9. L427: ‘revered suppression’ – change to ‘reveresed’?
10. L432-433: unclear as written – consider rephrasing
11. L444: perhaps the authors meant weakly instead of ‘weekly’
12. L529” ‘this nay’ – change to ‘may’
13. L612: change ‘drivrn’ to driven?
Comments on the Quality of English LanguageSome minor improvements to the language are suggested (under my minor points section).
Author Response
We appreciate your complementing comments “I found this to be a useful and well-organized review, with a complex topic presented in a simple and easy to understand manner. The authors cite most relevant work in the field and have included several figures and tables to summarize key information. I believe this review will be useful for readers looking to understand the role of Tregs in vaccine efficiency.”
In response to specific comments, we have made the following changes. As suggested about “vaccination vs. infection” to consider some discussion about live attenuated and inactivated vaccines for flu vaccines. Although the number of papers published about this specific subject is limited, we found some interestingly described issues. The additional paragraph has been included in new re-organized lines 221-235:
The same section includes the discussion on the effect of intranasal and intradermal administration on T cell response to vaccines.
All other specific comments have been addressed as requested, namely points 1 to 13 (note all lines have changes because of significant shortening of the manuscript and other editorial changes): as requested corrections were made for lines 1-4; line 5 correctly described efferent lymphatics, and lines 150-155 had been removed from the text. Remaining points have been corrected.
- L34: should read as ‘subsequent discovery of attenuated virus vaccine against polio”
- L61: change ‘plaid’ to ‘played’
- L67-68: there is a duplication of the sentence that needs to be corrected
- Figure 1: the legend should clarify the difference between the combinations of single and double – and + signs used to denote phenotypes.
- L90: should this read as ‘efferent’ instead of afferent?
- L150-155: as written these sentences are unclear to me – consider rephrasing to improve clarity
- L170: change ‘Th1-plarized’ (to polarized?)
- Section 3.4: some initial information defining the Tr35 subset more rigorously would be useful here.
- L427: ‘revered suppression’ – change to ‘reveresed’?
- L432-433: unclear as written – consider rephrasing
- L444: perhaps the authors meant weakly instead of ‘weekly’
- L529” ‘this nay’ – change to ‘may’
- L612: change ‘drivrn’ to driven?
Reviewer 2 Report
Comments and Suggestions for Authors
Stepowski et al present a review on the roles of Treg cells in the immune response, especially in the context of vaccination.
The topic is interesting and relevant. As the proportion of older people is growing in the industrial countries, it needs to be better understood how mechanisms that dampen responses to vaccines can be overcome. A vast amount of key literature is cited and the topic is covered broadly.
However, in its current form, the review is too long, very complex to read and in some parts not focused enough on the central topic. I recommend an intensive shortening of the text and of the numbers of figures.
Specific points:
- the introduction part on the T-cell response should be shortened. While it is certainly important to know that there are different subsets of memory and regulatory T cells, especially chapters 2 and 3.3 should be shortened in order to really focus on the key messages on the relevant Tregs in vaccination.
- On the other hand, chapters 6.1, 6.2. and 6.3 describe general ways of improving vaccination without a direct link to Treg cells. These chapters can be summarized in 1-2 sentences, citing the appropriate references.
- table 3 is not necessary as it only presents references.
In general, the figures should be re-designed. They are not always informative and not easily accessible:
- Figure 1 and Table 1 have redundancies and should be combined. The legend of Figure 1 should not repeat the information presented in the figure.
- Figure 5 contains too much text which is hardly readable.
- Figure 6: it is very hard to get information out of this drawing. If the figure is to be maintained, it needs re-design with the key-information that is easily accessible.
Minor points:
Line 35: peptide-based universal influenza vaccines can certainly not be named as a milestone like the polio vaccine. They are still in development, whereas the polio vaccines have saved millions of lives.
Lines 256-258: doubling of “vaccination of elderly”.
Line 262: the sentence does not work
Line 285: replace vaccines by vaccination
Line 302: check reference 103
Comments on the Quality of English LanguageThere are some typos and missing words throughout the manuscript. Please re-check.
Author Response
We appreciate positive comments “The topic is interesting and relevant. As the proportion of older people is growing in the industrial countries, it needs to be better understood how mechanisms that dampen responses to vaccines can be overcome. A vast amount of key literature is cited and the topic is covered broadly.”
The Introduction has been shortened, and especially chapters 2 and 3.3.
Chapters 6.1, 6.2 and 6.3 have been combined and limited to a few sentences.
Figure 1 and Table 1 have been combined, and Tables 1 and Table 3 were removed.
Figure 5 has been redesigned and some text removed or re-edited for better reading.
Figure 6 has been redesigned and text simplified for easy reading.
The reference 103 has been replaced with correct one.
All other points have been specifically addressed, but locations of changes have changed:
Line 35: peptide-based universal influenza vaccines can certainly not be named as a milestone like the polio vaccine. They are still in development, whereas the polio vaccines have saved millions of lives.
Lines 256-258: doubling of “vaccination of elderly”.
Line 262: the sentence does not work
Line 285: replace vaccines by vaccination
Line 302: check reference 103
Reviewer 3 Report
Comments and Suggestions for Authors
Dear Editor,
The article entitled: "The major role of T regulatory cells in efficiency of vaccination in general and immunocompromised populations: a review" by Stepkowski et al is a comprehensive review that aims to discuss about strategies that will improve the efficacy of vaccination in humans and especially in immunocompromised and older individuals as well as organ transplant patients. The review is very well written and to my opinion is suitable for publication after minor revision.
Major comments:
1. As mentioned in the title the review should focus only in immuncompromised patients but unfortunately there is a huge introduction and a part about the role of Treg cells in vaccinations. To my opinion this part should be shortened without loss of significance because is difficult for the reader to follow it.
2. Authors should divide patients with cancer in different subpopulation. I mean those who undergo intensive chemotherapy, those who undergo HSCT and additionally they could add a paragraph that focuses on different age group (children and adults). Are any differences in the immune response between immunocompromised children and adults after treatment for cancer?
Minor comments
1. Figure 5 is difficult to understand.
2. Authors should discuss about vaccinations and the efficacy of the immune system in the era of targeted therapies for cancer.
Author Response
We sincerely appreciate your positive comments “The review is very well written and to my opinion is suitable for publication after minor revision.” As requested, we have significantly shortened the introduction. The patients have been divided into those undergoing intensive chemotherapy and those who undergo HCCT and in different age groups. The section has been added to the manuscript between lines 327 and 380:
In addition, all minor comments have been addressed, namely Figure 5 has been completely redesigned and we have discussed about vaccination and the efficacy of the immune system in the era of targeted therapies for cancer.
- Figure 5 is difficult to understand.
- Authors should discuss about vaccinations and the efficacy of the immune system in the era of targeted therapies for cancer.
In response to Reviewer # 3:
We are grateful for the positive comments “The topic is interesting and relevant. As the proportion of older people is growing in the industrial countries, it needs to be better understood how mechanisms that dampen responses to vaccines can be overcome. A vast amount of key literature is cited and the topic is covered broadly.”
As requested, we have made the following changes:
The Introduction section has been shortened, and especially chapters 2 and 3.3.
Sections 6.1, 6.2 and 6.3 have been eliminated and changed into one major section with few sentences written about each subject.
Figure 1 and Table 1 have been combined as previous Table 1 has been removed.
Figure 5 has been redesigned and its text improved
Figure 6 has been redesigned and improved with only relevant text.
All other points have been addressed as requested, but lines have changed.
Round 2
Reviewer 2 Report
Comments and Suggestions for Authors
The authors have addressed most of the comments of the first review.
While some parts have become more focused now, a novel paragraph on CAR-T cell therapy has been added (lines 332-371). This is off topic, as the review is supposed to be on Tregs in vaccination. Besides, the roles of Tregs in cellular immunotherapy is such a complex issue on itself, that it cannot be addressed as a sub-chapter in sufficient scientific detail. I strongly recommend to remove this paragraph again.
The “response to the reviewer” is not clear and detailed enough. It must become clear in the text, what parts exactly were removed and/or changed. A tracked-changes version is mandatory.
Other points:
iTreg cells are now not introduced but appear in the text and table without explanation.
Figure 5 is still overcrowded with text, and I doubt whether the text will be readable in a print-out version of the paper.
Comments on the Quality of English LanguageMinor editing on grammar is necessary.
